# MAPK Pathways in Ocular Pathophysiology: Potential Therapeutic Drugs and Challenges

**DOI:** 10.3390/cells12040617

**Published:** 2023-02-14

**Authors:** Petros Moustardas, Daniel Aberdam, Neil Lagali

**Affiliations:** 1Division of Ophthalmology, Department of Biomedical and Clinical Sciences, Linköping University, 58185 Linköping, Sweden; 2INSERM U1138, Centre des Cordeliers, 75270 Paris, France; 3Université de Paris Cité, 75014 Paris, France; 4Department of Ophthalmology, Sørlandet Hospital Arendal, 4838 Arendal, Norway

**Keywords:** MAPK, ERK, p38, JNK, ocular diseases, therapy, eye toxicity

## Abstract

Mitogen-activated protein kinase (MAPK) pathways represent ubiquitous cellular signal transduction pathways that regulate all aspects of life and are frequently altered in disease. Once activated through phosphorylation, these MAPKs in turn phosphorylate and activate transcription factors present either in the cytoplasm or in the nucleus, leading to the expression of target genes and, as a consequence, they elicit various biological responses. The aim of this work is to provide a comprehensive review focusing on the roles of MAPK signaling pathways in ocular pathophysiology and the potential to influence these for the treatment of eye diseases. We summarize the current knowledge of identified MAPK-targeting compounds in the context of ocular diseases such as macular degeneration, cataract, glaucoma and keratopathy, but also in rare ocular diseases where the cell differentiation, proliferation or migration are defective. Potential therapeutic interventions are also discussed. Additionally, we discuss challenges in overcoming the reported eye toxicity of some MAPK inhibitors.

## 1. Introduction

The mitogen-activated protein kinase (MAPK or MAP kinase) family consists of protein kinases that phosphorylate their own dual serine (Ser) and threonine (Thr) residues (autophosphorylation), or those found on their substrate downstream kinases, to activate or de-activate their target [1]. MAPKs are ubiquitously expressed and evolutionarily conserved in eukaryotes. Each group of MAPKs contains a multi-tiered signaling cascade of kinases: at the top upstream level of the canonical MAPK pathways there are the MAPK kinase kinase kinases (MAPKKKKs, or MAP4Ks), which act upon MAPK kinase kinases (MAPKKK, or MAP3Ks), which then act in turn on MAPK kinases (MAPKKs, or MAP2Ks), with a final effector MAPK as their target (Figure 1). MAP3Ks are Ser/Thr protein kinases that are activated through phosphorylation., which, in turn, leads to the phosphorylation and activation of MAP2Ks in their Ser/Thr activation site (Ser-X-X-X-Ser/Thr motif). Activated MAP2Ks then stimulate MAPK activity through dual phosphorylation on Thr and Tyr residues within a conserved Thr-X-Tyr motif located in the activation loop of the MAPK domain [2] (Figure 1). A comprehensive list of MAPKs, MAP2Ks and MAPKs is presented in Table 1. MAPKs mainly include four subfamilies based on the conserved Thr-X-Tyr motif: ERK1/2, the JNK1/2/3, the p38 (α, β, γ, and δ), and the ERK5 branches, which are all ultimately activated by signaling cascades initiated by multiple factors such as growth factors and stress. More details on the signaling pathway members are given elsewhere [3]. Once activated through phosphorylation, these MAPKs in turn phosphorylate and activate an array of transcription factors present in the cytoplasm and nucleus, leading to the expression of target genes and resulting in a biological response. MAPKs are involved in multiple cellular processes, such as cell differentiation, proliferation, apoptosis, inflammation, stress responses, and immune defense [4]. In general, the activation of ERK by growth factors, hormones and proinflammatory stimuli promotes cell proliferation, whereas the activation of p38 and JNK by cellular and environmental stresses promotes multiple cellular processes such as proliferation, apoptosis, immunological effects, insulin signaling and neuronal activity [5]. The ERK pathway was the first MAPK cascade to be elucidated and is the best characterized. The canonical intracellular part of the activation pathway starts when a Ras GTPase exchanges a guanosine diphosphate (GDP) for a guanosine triphosphate (GTP) molecule [2]. This is facilitated upon the binding of extracellular mitogens to a cell surface receptor such as EGFR and the subsequent docking and activation of intracellular complexes, for instance GRB2-SOS [2]. This switching of Ras allows it to activate a MAP3K (e.g., Raf) and initiate the cascade of a MEK1/2 (MAP2Ks) and ERK1/2 (MAPK) activation (Figure 1). More generally, the ERK1/2 pathway is stimulated in mammalian cells by tyrosine kinase receptors and G-protein-coupled receptors through both Ras-dependent and Ras-independent pathways [6]. ERK1/2 is also activated by growth factors, mitogens, cytokines, osmotic stress, and in response to insulin [2]. Given its central role in cell proliferation, differentiation and survival, the MAPK pathway network and its inhibition has attracted great pharmacological interest in the context of cancer research, and a plethora of compounds have been developed/identified that can directly act on this pathway to influence cell fate.

The eye is the central organ of the visual system of animals, allowing vision and other, vision-independent photo-response functions by collecting light from the environment and converting the light information to neuronal impulses, ultimately terminating in the brain’s visual cortex. In vertebrates, this is achieved via a complex system of structures which are organized in a spherical organ and serve distinct roles. Figure 2 presents an overview of key events, structures and cell lineages during the early development of the eye in mammals. The developed eye structures of relevance in this review are illustrated in Figure 3. Briefly, from anterior to posterior, they include: the ***cornea***, a transparent, multilayered, circular convex structure in the front of the eye, through which light passes and is refracted as a first part of a light focusing mechanism; the ***limbus***, which is the transition area between the cornea and the sclera (that surrounds the rest of the eyeball), where limbal stem cells reside providing a barrier against vascularization in the cornea, maintaining a distinct immunocompetent environment and regenerating the corneal epithelium; the ***uvea***, which is the vascular middle layer of the eye, containing the choroid, the ciliary body and the iris; the ***iris***, a thin, pigmented, annular structure that forms a central pupil through which light passes, controls the pupil’s diameter and size by dilating or constricting, and thus regulates the amount of light that passes through; the ***ciliary body***, a ring-shaped thickening that supports the lens in place via the zonular fibers, controls its shape with the ciliary muscle, and divides the posterior chamber from the vitreous body; the ***lens***, a transparent biconvex structure located posterior to the iris, which can change shape to accommodate precise focusing of the light to the retina; the ***anterior chamber***, the space between the cornea, iris and the lens, filled with the aqueous humor; the ***posterior chamber***, a narrow space behind the peripheral part of the iris, and in front of the ciliary body and the lens suspending ligaments; the ***trabecular meshwork***, a spongy area around the base of the cornea and near the ciliary body, responsible for draining the aqueous humor from the eye via the anterior chamber; the ***retina***, which is a multilayered structure at the back and inner side of the eye, responsible for detecting light and converting it into neuronal impulses and consisting of the ***neural retina***, with layers of photoreceptors, interconnected neurons and supporting Müller glial cells, and the ***retina pigmented epithelium*** (RPE), a layer of pigmented cells which is located beneath the neural retina, supporting and nourishing it, maintaining an inner environment of “immune privilege” and absorbing scattered light; the ***macula***, an oval-shaped pigmented area in the center of the retina with high photoreceptor density and responsible for high-resolution, color vision; the ***fovea***, a pit at the center of the macula with closely packed cones that is responsible for sharp central vision, and the ***optic nerve***, which exits the eye through the optic disk in the retina, carrying the optical information to the brain.

This review focuses on the roles of MAPK signaling pathways in ocular pathophysiology and the potential to influence these for the treatment of eye disease. The aim of this review is to summarize the current knowledge of the identified MAPK-targeting compounds in the context of ocular diseases such as macular degeneration, cataract, glaucoma and keratopathy, but also rare ocular diseases where cell differentiation, proliferation or migration are defective. Additionally, challenges in overcoming the reported eye toxicity of some MAPK inhibitors will be discussed.

## 2. Physiological Role of MAPKs in the Eye

MAPK/ERK signaling, as a master proliferation and cellular differentiation regulation pathway, is indispensable for the formation of the organism as a whole during development [7]. More precisely, ERK kinases play important roles in promoting embryonic survival and regulate the development of the eye in vertebrates. Of note, although the process depicted in Figure 2 is largely conserved within vertebrates, fish such as zebrafish do not form a lens pit and vesicle; rather, the cells from the lens placode proliferate and migrate inwards, directly forming a solid spherical mass that detaches from the surface ectoderm. The formation of the neural retina, RPE and the cornea follow the same process and lineage. In adult goldfish, ERKs are highly expressed in multiple ocular tissues including the lens epithelial cells, lens fiber cells and the retina, whereas its inhibition promotes early apoptosis, preventing the formation of the eye [8]. Underscoring the importance of the ERK pathway in development, all RASopathies, which are pathologies due to mutations in the RAS-MAPK pathway, are confined to only gain-of-function mutational defects that lead to inefficient inhibition of the pathway, while there is no documented RASopathy caused by mutational pathway knockout [9]; since such mutations should be more common than gain-of-function mutations, their absence signifies that when they occur, are most likely non-viable. Regarding ocular development, morphology and function, RASopathies present only minor clinical manifestations such as the appearance of Lisch nodules, which are aggregates of dendritic melanocytes forming papules in the iris [10]. Given its importance for cellular functions, MAPK/ERK signaling has been implicated in multiple organisms in the processes of wound healing and regeneration. For instance, ERK2 is essential for retinal pigment epithelium (RPE) cell proliferation in vitro [11,12]. Although in mammals, the RPE is post-mitotic in the adult, the mechanisms underlying RPE proliferation are important for stem cell applications and for developmental understanding. MEK–ERK signaling is strengthened by auto-regulation of the expression of constituent molecules in the pathway [13], but blockade of initial MEK–ERK signaling inhibits the cell-cycle re-entry of newt RPE cells [14], and after wounding in the adult newt [15]. The MEK pathway is also essential to switch adult newt RPE cells to neural cells. [16]. Regeneration of a complete neural retina can be achieved in larval Xenopus Leavis through the activation of the MAPK signaling pathway by administering exogenous FGF-2 [17]. In zebrafish, retina regeneration after injury depends on Müller glia (MG) dedifferentiation into a cycling population of multipotent progenitors via an EGFR/MAPK signal transduction cascade that regulates the expression of regeneration-associated genes such as PAX6 [18,19]. It should be noted, however, that mammals, unlike teleost fish, do not possess the innate ability for retinal regeneration; rather, mammals develop gliosis after retinal damage. Thus, this knowledge is relevant to humans in the context of stem cell research, the potential for interventions to induce regeneration, or in developmental research. During rat embryogenesis, the ERK1/2 pathway is required for the proper development of retino-geniculate connections [20]. FGF2 stimulates PAX6 expression during the induction of transdifferentiation of the RPE through a FGFR/MEK/ERK signaling cascade into a neural-like epithelium [21]. Similar transdifferentiation is obtained in chicks through the ectopic expression of a constitutively-activated allele of MEK-1 [22]. In the injured chick retina, the MG showed an accumulation of p-ERK1/2 [23]. Regarding the JNK activation pathway, the upstream kinases MKK4 and MKK7 have redundant and unique roles in molecular signaling that are important for retinal development, RGC maturation and the response to axonal injury signaling [24]. JNK and p38 phosphorylation is increased after retinal ischemia, mainly in amacrine, ganglion and bipolar cells while ERK is activated in MG cells [25]. Specific blockage of ERK and p38 phosphorylation, but not of JNK, prevents ischemia-induced apoptosis and improves retinal function in a rat model [25]. Other studies have demonstrated, for instance, that in vivo inhibition of p38 MAPK activity may be detrimental to injured photoreceptor cells [26]. Thus, the use of p38 MAPK inhibitors for therapeutic purposes must take into account the possible side effects. p38 is activated in retinal ganglion cells (RGCs) after optic nerve axotomy, and this activation is in the signaling pathway for RGC apoptosis [27]. MAPK also plays a significant role in MG cell proliferation and differentiation within the retina, in a stage-dependent manner. Prior work strongly supports a model whereby activation of the MAPK signaling pathway promotes the entry of progenitors into a MG cell differentiation pathway during embryonic retinal development, but not after birth [28]. For example, Shp2 protein phosphatase deletion abolished ERK phosphorylation in the neural retina, leading to extensive retinal cell death and degeneration. Additionally, Shp2 mediated a basal level of Ras-MAPK signaling in MG cells during postnatal development and in an adult retina under normal physiological conditions [29]. Also, the ERK1/2 and p338 MAPK pathways are key regulators of growth cone guidance in vitro [30].

## 3. Link between the MAPK Pathway and Ocular Pathophysiology

Age-related macular degeneration (AMD), diabetic retinopathy (DR), retinal dystrophies (RD) and glaucoma are the leading neurodegenerative diseases in the human visual system, eventually leading to vision loss. RD and AMD are characterized by the loss of photoreceptor cells, while glaucoma results in the loss of RGCs. As described below, MAPK signaling is involved in the pathogenesis of these ocular diseases. Figure 3 presents a synopsis of the described ocular structures and conditions, in which MAPK interactions are discussed.

### 3.1. MAPK in the Retina

***Age-related macular degeneration (AMD):*** AMD is a major cause of central visual loss. Dry AMD is characterized by the progressive loss of the photoreceptor cell population in the retina while the “wet” or “neovascular” form of AMD is characterized by the growth of new abnormal blood vessels from the choroid into the retina, associated with rapid and severe vision loss. As detailed below, the involvement of MAPK signaling in AMD has been studied and showcased in multiple contexts, such as in the effects on RPE cells in in vitro models of oxidative stress and UV-induced damage, in wet AMD neovascularization, in mouse models that partly recapitulate aspects of the disease, in GWAS studies, in the context of inflammation and in studies regarding the pharmacologic pathway inhibition. Using bioinformatics and GWAS tools, several MAPK pathway genes, including the JNK signaling pathway, have been identified as targets for an increased risk of advanced AMD pathology [31] and the activation of ERKs has been associated in RPE-choroid AMD phenotypes [32]. A hint for a potential mechanistic involvement of MAPKs in AMD comes from the finding that the RPE of patients with geographic atrophy, an untreatable advanced form of dry (non-neovascular) AMD, have reduced miRNA processing enzyme DICER1, and the ablation of this specific gene is sufficient to induce RPE degeneration in mice [33] by causing toxic levels of RNA from the Alu family of short interspersed elements in the genome and specifically promoting the phosphorylation of ERK1/2 in vivo [34]. In a different mouse model of wet AMD, phosphorylated ERK1/2, JNK, and p38 were all found to be upregulated [35], while in yet another mouse model of laser-induced choroidal neovascularization, JNK inhibition led to a decrease in apoptosis, decrease in VEGF expression, and a reduction in neovascularization [36]. Most data come from RPE cell cultures, as RPE cells play multiple roles in the ocular system by interacting with photoreceptors and maintaining ocular homeostasis [37]. Furthermore, RPE dysfunction causes diseases related to vision loss, such as AMD. UV-C irradiation of RPE cells induces cell death through the activation of both JNK and p38 that is reversed by the use of specific kinase inhibitors, namely, D-JNKi and SB203580 [38]. Two studies have reported the anti-oxidant effects of the green tea extract, EGCG, in protecting RPE cells from UVA-induced damage and apoptosis while inhibiting the UVA-induced ERK activation [39], and they implicate an “anti-apoptotic role” of JNK activation in this context [40]. It has also been reported that ERK1/2 phosphorylation can occur within 15 min of RPE cell exposure to cigarette smoking extract, leading to increased autophagy [41], which is a cellular process recognized to play a role in AMD [42]. The MEK/ERK pathway and not the JNK pathway have also been found to be involved in oxidative stress-induced apoptosis in RPE cells, with sustained MEK/ERK pathway inhibition rescuing RPE cells from cell death [43]. Another study has shown that PD98059, a potent and selective inhibitor of MEK, inhibits ERK1/2 phosphorylation and blocks RPE degeneration both in RPE cell cultures and in mice [34]. These effects seem to be dose-dependent, however, and reliant on a non-total ERK1/2 inhibition, since specific ablation of ERK1/2 in mouse RPE cells leads to RPE cell death and retinal degeneration [44]. Multiple MEK pathways have been also reported to be involved in retinal ischemia, with each demonstrating a specific pattern of cellular expression, while the blockade of p38 or ERK can provide significant protection from ischemic damage [25]. VEGF is known to be the major angiogenic factor involved in AMD [45], and increased VEGF-A levels are causally implicated in the progressive dysfunction of the RPE [46]. Under oxidative stress conditions, induced RPE cell apoptosis has been shown to be ameliorated via the ERK/p38/NFκB/VEGF signaling pathways [47]. The precise mechanism in the connection between MAPK signaling and oxidative stress-induced RPE apoptosis is, however, complex. Constitutive and oxidative-stress-induced VEGF expression in the RPE are differently regulated by different MAPKs, with only p38 being necessary for a baseline VEGF expression, while ERK shares an active role with p38 under stimulated overexpression [48]. The effects of ERK in the balance between cell death and survival under oxidative stress seem to also be time-dependent, with a biphasic ERK1/2 signaling in the RPE where only late ERK1/2 activation is pro-apoptotic [49]. Generally, MAPK pathways play an essential role in modulating VEGF. Constitutive activation of ERK1/2 leads to increased VEGF expression [50], while overexpression of p38 and JNK also leads to VEGF expression elevation [51]. In vitro, ERK1/2 modulate the release of VEGF from retinal ganglion cells [52]. Conversely, under oxidative stress-signaling conditions, VEGF itself activates MAPK pathways [53]. Nutritional anti-VEGF intervention against wet AMD has been shown to act by preventing MAPK activation in RPE cells [54], with the involvement of NFκB [55], and similar effects have been demonstrated in mouse models [56]. Transforming growth factor β (TGFβ) signaling deficiency in retinal neurons and MG cells mediates a shift in the expression of MAPK signaling pathway regulators from pro-survival to pro-apoptosis [57]. Conversely, the stimulation of TGFβ signaling or the activation of pro-survival MAPK signaling pathways in retinal neurons or in MG cells decreases the degeneration of photoreceptors in diseases such as RD or AMD [57].

On the other hand, the MEK signaling pathway is also involved in serum-induced RPE cell proliferation [11]. The same pro-proliferative effect has been demonstrated after hydrogen peroxide stimulation, where ERK1/2 does not seem to be involved in cell death, but rather with RPE proliferation [58]. In retinal ganglion cells, all three branches of the MAPK cascade have been reported to be protective against hydrogen peroxide-induced apoptosis [59]. As it has also been shown that JNK and p38 activation is important for hydrogen peroxide-induced apoptosis in RPE cells [60], it seems that this dual role is dependent on the signal duration, with short-term ERK1/2 signaling being pro-proliferative and, inversely, a persistent ERK1/2 activation leading to cell death [43,61,62].

A link between AMD, inflammation and MAPK has been reported as well. Exposure of the retina to excessive light (i.e., 10,000 lux for 30–60 min in a photosensitive Abca4^−/−^Rdh8^−/−^ mouse model) causes photoreceptor cell death, retinal inflammation and other degenerative changes [63]. MAPK pathways participate in the processes underlying this light-induced photoreceptor degeneration. In a rat model of light-induced retinal degeneration, ERK1/2, p38 and JNK were phosphorylated and activated, mainly in the retinal outer nuclear layer (ONL) [64]. In this model though, only the p-ERK1/2 inhibitor attenuated the progressive photoreceptor loss and thinning of the ONL and protected the retina from photoreceptor degeneration and inflammation, while p38 and JNK inhibitors had no such effect. Inflammation is a key factor in RPE dysfunction in retinal degenerative diseases. MAPK signaling pathways are the most widely used in initiating the release of cytokines and chemokines in RPE cells [65]. High serum levels of IL-6 correlate with the development of AMD [66], and the expression of IL-6 seems to depend partly on p38 activation, which is initiated itself through a decline of proteasome activity which gradually decreases in RPE cells during AMD [67]. The pro-inflammatory cytokine IL1β is also highly activated in AMD [68]. The anti-inflammatory natural compound, quercetin, was reported to protect RPE by reducing IL1β production and, as a consequence, reducing the phosphorylation of MAPKs (i.e., ERK1/2, p38 and JNK1/2) and their pro-inflammatory targets such as IL-6 and IL-8 [69].

***Diabetic retinopathy (DR)****:* Ocular involvement in diabetes occurs early in the course of the disease and is present in over a third of type 2 diabetes mellitus patients at the time of diagnosis. Blindness due to diabetic retinopathy (DR) remains a leading cause of adult-onset blindness. Diabetes has been found to activate p38 MAPK in a variety of tissues, including the kidneys, nerves, vasculature, and heart [70]. p38 activation also plays a role in the long-term vascular pathology of diabetic retinopathy while its inhibition by PHA666859 in an STZ-induced diabetes rat model blocked the death of endothelial cells and pericytes and significantly inhibited the degeneration of retinal capillaries [71]. The phosphorylation levels of p38 have been associated with the viability of high-glucose-treated RF/6A cells and their in vivo viability in a STZ diabetic rat model [72], and inhibition of p38 protected against diabetes-induced retinal microvascular damage in rats by a mechanism involving NFκB [73]. Inhibiting the p38-MAPK pathway can also prevent retinal neurodegeneration [74]. In a mouse model of DR, p38 phosphorylation was activated by corticotropin-releasing hormone activation, the inhibition of which reduced the murine visual impairment [75].

A pro-survival effect of ERK1/2 has been reported from in vitro studies modeling the early phases of DR, where MG activation was suggested to have a neuroprotective activity against high-glucose-induced neurotoxicity via this pathway [76]. Additionally, MG are primarily responsive to intra-ocular insulin with increased phosphorylation of ERK1/2 and p38 [77]. In vitro, VEGF reduction by dimethyl fumarate, accompanied by p38 signaling inhibition in human RPE cells attenuated high glucose-induced apoptosis [78]. These findings have been corroborated by other studies that show that in the retina of STZ-induced diabetic rats, pERK1/2 and VEGF are approximately synchronized, and that the protein level of VEGF is regulated by ERK1/2 phosphorylation. These data suggest that VEGF, Ets-1, and ERK1 act synergistically in the development of DR [79]. Other in vivo studies have also concluded that the ERK1/2 signaling pathway is involved in VEGF release in the diabetic rat retina [80], and that it specifically regulates VEGF secretion by MG cells [81]. Thus, it seems that ERK1/2 participates in a sensitive balance between neuroprotection on one hand, and increased vascularization on the other. Similarly, the ERK1/2-VEGF signaling pathway was reported to be activated following the treatment of Müller cells with diabetic blood isolated from rats treated with traditional Chinese small compounds known to promote blood circulation. Such treatment improved the diabetic rat Müller cell metabolism [82,83].

***Retinal dystrophies (RD)****:* Retinal dystrophies are a heterogeneous group of hereditary diseases that cause progressive and severe loss of vision by altering the anatomy and/or function of the retina. The most common of these dystrophies is retinitis pigmentosa (RP), which is a group of inherited, phenotypically and genetically variable disorders, characterized by progressive dysfunction, cell loss, and eventual atrophy of retinal tissue affecting 1 in 4000 people worldwide [84]. RP is typically caused by mutations in key photoreceptor genes, with over 50 genes known to be affected [85]. A heterozygous dominant mutation in MAPKAPK3 (a downstream substrate of p38) is responsible for a specific inherited human retinal pigment dystrophy with severe defective RPE, that has been confirmed in a mouse model [86]. At the same time, a balanced p38/TXNIP/NFκB pathway inhibition is required for proper tight junctions in the inner blood–retinal barrier [87] and for the prevention of macular edema [88,89,90,91], while ERK and JNK phosphorylation is inversely correlated with the expression of tight junction components in human retinal endothelial cells in vitro [92]. Additionally, a large body of evidence points towards an indispensable role of MAPK pathways in neuroprotective gliosis. ERK signaling suppression is apparently crucial for MG cell survival [93], since ERK1/2 was activated in a mouse model with atypical gliosis and retinal degeneration [94]. Protective effects on photoreceptor degeneration have also been shown after the inhibition of ERK1/2 activation by PD98059 in Müller cells [95]. In rats with chronic ocular hypertension, Kir channel inhibition-induced Müller cell gliosis was mediated by the MEK-ERK/p38-CREB/c-fos signaling pathway [96]. Moreover, the ERK1/2 and PI3K/AKT pathways in Müller cells have been identified as mediators for the effects of NGF on gliosis and also for VEGF-mediated angiogenesis [97].

***Retinal injury—neuroprotection:*** Although inhibition of p38 or ERK has been shown to confer significant protection from ischemic damage as assessed by retinal cell layer thickness and apoptosis measurement [25], seemingly conflicting effects of p38 signaling having neuroprotective action have been also reported in retinal ischemia/reperfusion injury [98]. A fascinating phenomenon regarding retinal blood perfusion that could perhaps be part of this apparent discrepancy is ischemic pre-conditioning, where a brief ischemic episode can confer protection against retinal ischemic injury in rats [99]. Retinal ischemia-reperfusion increases the expression of ERK1/2 in the neuroretina and retinal arteries [100], and depending on ischemia duration, this effect could conceivably be part of pre-conditioning, priming cells to better respond to future ischemic incidents. p38 regulation by MKP-1 has also been found to be a pathway involved in neuroprotective ischemic pre-conditioning [101], which was lost after the specific inhibition of p38α [102].

In N-methyl-D-aspartate (NMDA)-induced excitotoxic damage of retinal ganglion cells, a standard model of retinal injury, JNK and p38 have been found to be proapoptotic in the retinal ganglion cell layer, with c-Jun synthesis and phosphorylation participating in NMDA-induced neuronal cell death [103], but ERK has not been found to have similar effects [104]. In the same fashion, another study concluded that JUN can facilitate signaling events triggered by axonal injury that ultimately culminate in RGC apoptosis. [105]. Increased phosphorylation levels of ERK1/-2 and Akt, as well as a decreased caspase-3 activity, however, were observed in injured mouse retinae following optic nerve transection in vivo, but they were described to confer protection against degeneration [106]. Additionally, MAPK and PI3K-Akt pathways have been reported as necessary for neuroprotective signaling in axotomized retinal ganglion cells [107]. Injured photoreceptors generate pro-survival signals that induce protective mechanisms to rescue photoreceptors from apoptosis [26]. This could, however, be a double-edged sword, depending on the extent of the injury or stressor as, optimally, pro-survival signaling would be beneficial only to sub-critical injury incidents. Such is the case for leukemia inhibitory factor (LIF), which is produced by a subset of MG cells and was dependent on p38 in a light-induced model of retinal degeneration [26].

***Glaucoma:*** Glaucoma is an optic nerve degenerative disease, characterized by a pathologically-high intraocular pressure, RGCs loss, and visual function damage. It has been suggested that MAPK signaling pathways are involved in the onset or progression of glaucoma [108]. In a model simulating glaucoma conditions with biaxial stretching of optic nerve head astrocytes, bioinformatic and enrichment analysis identified the MAPK pathway, among others, to be of potential interest in cellular responses after mechanical stimulation [109]. With TGF-β stimulation, a factor known to be implicated in primary open angle glaucoma [110], human trabecular meshwork cells (TMCs) were transcriptomically analyzed and the MAPK cascade was found to be significantly enriched within the differentially expressed genes [111]. Similarly, the MAPK network has been found to be differentially expressed after glucocorticoid treatment in both glucocorticoid responsive and unresponsive TMCs [112]. Apart from bioinformatic enrichment evidence, in an experimental setting of chronic ocular hypertension induction via laser-induced coagulation of the trabecular meshwork in rats, ERK1/2, JNK and p38 exhibited a specific spatio-temporal expression pattern and also a distinct activation pattern in the retina, ONH and optic nerve [113]. Similarly, p38 activation in the retina with elevated IOP has been reported in rats and mice [114]. In another IOP elevation model in rats using mechanical compression, the JNK inhibitor SP600125 significantly promoted RGC survival [115]. Additionally, Jun deficiency has been shown to protect RGC somas from ocular hypertensive injury, but not RGC axons from glaucomatous neurodegeneration in mice, suggesting that JNK–JUN signaling plays a role as a pro-death signaling pathway between axonal injury and somal degeneration [116].

Regarding in vitro data in TMCs, TGF-β induces the expression of IL-6 in a p38-dependent manner that can be attenuated using the selective p38 inhibitor SB202190 [117]. Similarly, p38 has been reported to mediate the expression of extracellular matrix-related proteins in TGF-β stimulated TMCs. Collagen Type I upregulation has been shown to be suppressed by the p38 inhibitor SB203580 [118]. Additionally, the SPARC protein, whose expression inhibition in TMCs or knockout in mice has been associated with enhanced aqueous drainage [119,120], has been shown to be upregulated upon TGF-β stimulation, in a p38- and JNK-dependent manner, with an inhibition of either pathway being sufficient to suppress SPARC upregulation [121]. ERK1/2 has been shown to be modulated in a TGF-β2 dependent manner in concert with changes in extracellular matrix elasticity [122] and the enhancement of cellular adhesion strength [123]. TMC adhesion has been shown to be causally connected to ERK1/2, which directly affects the expression of molecules such as fibronectin, under various stimulatory conditions not necessarily involving TGFβ [124].

Retinal microglia are activated in the early phase of the glaucoma process [125]. Pharmacological inhibition of the p38 pathway reduced the pro-inflammatory activation of retinal microglia [126]. Meanwhile, ERK signaling negatively regulated the Rho-kinase-mediated human trabecular meshwork cell contractility, a feature related to primary open-angle glaucoma [127]. Additionally, a link between p38 and ERK1/2 with an excess of pro-fibrotic ECM production in lamina cribrosa, a key site of fibrotic damage in glaucomatous optic neuropathy, has been reported in glaucoma [128]. Of potential therapeutic interest, topical ocular delivery of the p38 inhibitor BIRB 796 showed a neuroprotective effect in animal models of glaucoma, although the intraocular pressure was not affected [129]. On the other side of the spectrum, haploinsufficiency of JNK2, or both JNK2 and JNK3, results in an increase in ocular hypertension-induced neurodegeneration, suggesting JNK2 as a potential tool for glaucoma treatment [130].

### 3.2. MAPK in the Cornea

The cornea, as the most anterior tissue of the eye globe, is directly exposed to the outer environment, exposing it to potential traumatic damage. The healing process of corneal wounds involves cell apoptosis, migration, proliferation, differentiation, and extracellular matrix (ECM) remodeling.

***Corneal wound healing**:* Like all other epithelial barriers, the corneal epithelium is constantly subjected to the outer environment, with a constant exposure to mechanical, chemical, and biological insults. It can, thus, be said that it is in a constant phase of “healing”, a process that involves the proliferation of basal epithelial cells, the proliferation and centripetal migration of the limbal cells and epithelial cell loss from the surface [131]. But even in more acute injuries, the corneal epithelium responds rapidly by similar migratory mechanisms via activation of the EGF family, KGF, HGF, IGF, insulin, and TGF-β to cover the epithelial defect and to re-establish its barrier function [132]. It naturally follows that the MAPK cascade, which is responsive to the above growth factors and regulates proliferation and cell fate, is crucially implicated in the wound healing process. In epithelial cell wound healing in general, and not confined to the corneal tissue, the ERK-mediated c-Jun N-terminal kinase regulates lamellipodial protrusion in migrating epithelial cells and cell sheets during epithelial wound closure [133]. During corneal wound healing, ERK1/2 is activated and plays an important role in the initiation of cell migration and proliferation [134]. Both the p38 and ERK signaling pathways play a role in promoting cell migration of the corneal epithelium during the healing process of corneal injury [135], in a phosphorylation crosstalk that balances between increased cell migration and attenuation of proliferation at the leading edge of the wound [136,137,138]. These processes are orchestrated mainly through TGF-β signaling [138,139,140,141] and the temporal/spatial distribution of its isoforms [142] or EGF stimulation [136]. In pharmacologic intervention studies, the ERK pathway has been identified as a mediator of the enhanced corneal epithelial wound healing process after stimulation by various compounds such as Sericin [143] or Diquafosol [144] in rat models and human cells in vitro [143], and Chitosan [145] and diadenosine polyphosphates [146] in rabbit primary corneal epithelial cell models. Moreover, the activation of ERK1/2 is responsible for the disruption of epithelial tight junctions and, thus, the barrier function [147], which also is part of the natural epithelial cell loss and recycling, along with the reassembly of cellular adhesion structures during healing. In addition, TGF-β and basic-fibroblast growth factor (FGF-2) synergistically activate the p38 signaling pathway, inhibiting proliferation and promoting the migration of corneal endothelial cell injury repair in cultured corneal endothelial cells [148]. Indeed, p38 inhibition has been shown to be beneficial in the successful maintenance of human corneal endothelial cell cultures, enhancing proliferation and allowing colony expansion [149] while minimizing cellular senescence [150]. Likewise for the epithelium, a study that focused on tissue-engineering of corneal epithelium found that downregulation of p38 signaling helped maintain the self-renewal of limbal stem cells and prevented the terminal differentiation of corneal epithelial cells [151]. JNK signaling also contributes to injury-induced corneal epithelial migration [152], and it has been shown that JNK, together with p38, promote SPARC-dependent corneal epithelial wound healing [153], with the JNK pathway playing a crucial role in the regulation of scar formation in the resolution of corneal stromal trauma [154]. In other aspects of corneal wound healing, the MAPK pathway has been found to regulate NGF via ΔNp63, promoting the innervation of the cornea and epithelial homeostasis [155], and it has been found to also control the constitutive synthesis of collagenase-1 during fibrotic repair in the corneal stroma [156], stimulated by IL1-α.

***Dry eye****:* Dry eye disease (DED) is an inflammatory, ocular multifactorial disease of the tear fluid, secretion apparatus and ocular surface that results in potential damage of the ocular surface homeostasis. Inflammation plays a prominent role in DED pathogenesis. The expression of various proinflammatory cytokines and chemokines along with increased phosphorylated p38 and JNK [157], and a sustained activation of the three MAPK signaling pathways (i.e., JNK, ERK and p38) were found in a dry eye mouse model [158]. In another mouse study, desiccating stress significantly increased the expression of IL-1alpha, IL-1beta and TNF-alpha mRNA and stimulated the phosphorylation of JNK1/2, ERK1/2 and p38 MAPKs in the corneal epithelium, effects ameliorated by doxycycline [159]. The inhibition of phosphorylation of the p38 downstream substrate MAPKAPK2 reduced inflammation and ocular surface damage in a DED mouse model [160]. In a rabbit DED model, the Chinese plant, Esculetin, reduced inflammation and dry eye symptoms through specific inhibition of the ERK1/2 pathway [161], and recently it has been reported that an oral administration of Dendrobium officinale extract downregulated the phosphorylation of ERK in human corneal keratocytes and upregulated various aquaporins, while its oral administration in rats enhanced tear production and had a protective effect on ocular surface damage [162]. Accordingly, the levels of the three MAPK signaling pathways and inflammation were rapidly increased in a mouse DED model and remain elevated after 5 days [158]. Recent studies of DED have shown that autophagy activation can protect the ocular surface from inflammation via the inhibition of p38 [163].

***Keratoconus:*** Currently, only limited evidence for the involvement of MAPKs in keratoconus have arisen from recent studies of high-throughput proteomic and RNA-seq analyses and from bioinformatic enrichment studies. Transcriptional profiling of corneal stromal cells derived from patients with keratoconus found the MAPK pathways significantly enriched [164]. This was also confirmed in a multi-level combined gene enrichment meta-analysis regarding transcriptomic data from keratoconus-related research [165]. In a direct transcriptomic study focusing on mRNA and miR expression in epithelial and stromal cells from keratoconus patients, the MAPK pathway family was one of the most significantly enriched. Clearly, more and targeted research is required to validate these findings and to expand on the role of MAPKs in keratoconus.

***Aniridia****:* Aniridia is a genetic panocular eye disease caused by inherited or spontaneous mutations of PAX6, which lead to reduced protein levels, that are insufficient for normal eye development and function. Apart from obvious structural developmental defects such as an absence of the iris or iris coloboma, characterized by an abnormal and underdeveloped iris, a heterozygous deficiency in PAX6 (haploinsufficiency) may also lead to increased intraocular pressure and glaucoma, optic nerve hypoplasia and retinal detachments [166], and also in some cases iris-cornea adhesions and abnormalities of the posterior corneal stroma, Descemet’s membrane, corneal endothelium, the lens and anterior chamber, which are collectively called Peters’ anomaly [167]. Even in the absence of such congenital defects, aniridia is commonly accompanied by progressive ocular surface disease characterized by a progressive corneal opacification, called aniridia-associated keratopathy (AAK), also named aniridia-related keratopathy (ARK). AAK is associated with limbal stem cell functional breakdown, an ingrowth of blood vessels into the cornea, the sporadic appearance of goblet cells, and with inflammatory cell infiltration [168]. Corneal epithelial wound healing in a mouse model of AAK is delayed, in part due to defective EGF and calcium signaling, whereas addition of EGF prior to wounding rescues the calcium response in mutant PAX6^+/−^ cells and enhances ERK1/2 phosphorylation, restoring a rapid wound-induced cell migration [169]. The MAPK signaling pathway was shown to be enhanced in human aniridia conjunctival cells [170] and in vivo in PAX6^+/−^ mouse epithelia [171]. An in vivo partial rescue of the aniridia phenotype was obtained in an AAK mouse model by either a topical or oral administration treatment with the MEK inhibitor PD0325901, which restored normal Pax6 protein levels [172]. Beneficial effects of this inhibitor in PAX6^+/−^ mice in terms of an increased retina size and thickness, a reduced intraocular pressure and reduced retinal hotspots appears to be prolonged one year after postnatal treatment [173], although the effects on the cornea are currently unknown. However, as discussed below, the usage of broad spectrum compounds such as MEK inhibitors for clinical applications could lead to undesirable side effects, unless applied topically. Recently, a cellular model of Pax6 haploinsufficiency has been produced by genome editing on immortalized human limbal epithelial cells. This in vitro AAK model exhibited altered cellular functions which were corrected by exogenous Pax6 recombinant protein [174]; however, direct exogenous delivery of the protein in the correct dosage would be difficult in humans, as too little protein would be ineffective while too much could have detrimental effects. This cellular model was, therefore, subsequently used to screen for drugs able to rescue Pax6 by increasing its endogenous production in cells. Among the selected hits, Duloxetine and Ritanserin, two anti-psychotropic small compounds, were identified in the in vitro AAK cellular model as being able to restore PAX6 gene expression and limbal cell functional properties [175,176]. Of interest, both drugs re-activated PAX6 gene expression through specific MEK and ERK inhibition, respectively [175,176]. This was supported by the finding that PAX6 gene expression is down-regulated by EGF to allow for proper corneal epithelial proliferation, while the overexpression of PAX6 in these cells attenuated EGF-induced proliferation [177]. Although Ritanserin has not been marketed for medical use because of safety concerns, specific approved drugs such as Duloxetine could conceivably be translated to clinics relatively quickly, as pharmacovigilance and safety studies for oral administration in fibromyalgia and other patients prescribed with the drug did not show severe eye toxicity [178], apart from instances of blurred vision (6.7%) [179], uncommonly, diplopia, dry eye, mydriasis or visual impairment (<1%) and rarely, glaucoma (<0.1%) [180,181]. The discovery of already-approved small compounds that are able to enhance endogenous Pax6 activity, which could be locally administered using eye drops, may lead to the rapid development of applicable drugs for the topical treatment of keratopathy in aniridia.

### 3.3. MAPK in Other Ocular Pathophysiologies

Apart from pathologic manifestations in the retina, the cornea or the optic nerve, MAPK signaling is implicated in ocular conditions that involve other parts of the eye. Glaucoma, for example, which has been described previously, may also involve other structural parts of the eye such as the trabecular meshwork. Other ocular structures with MAPK interactions include the uvea and the lens. Below we describe these interactions and summarize the literature concerning them.

***Uveitis****:* Uveitis represents eye inflammation which damages mainly the uvea part of the eye (that includes the iris, ciliary body and choroid). Uveal melanocytes release MCP-1, IL-6 and IL-8 after an inflammatory stimulation by LPS or IL-1β that is accompanied by JNK1/2 upregulation and can also be attenuated by p38 inhibition [182,183]. It has been shown that human retinal endothelial cells also respond to LPS in vitro with increased p38 phosphorylation [184], and induction of experimental uveitis by LPS in vivo activated ERK on MG cells which was resolved by dexamethasone treatment [185]. Regarding the p38 pathway, its suppression has been correlated with anti-inflammatory effects [186]. Interfering with the p38 pathway by the use of the specific inhibitor XG-102 has been shown to reduce intraocular inflammation [187] in a dose-dependent manner [188] in a rat LPS-induced uveitis model and also in patients with post-operative inflammation after anterior and posterior segment combined surgery, glaucoma surgery or complex posterior segment surgery [189,190,191]. Apart from p38, inhibitors of other MAPKs, when used in the context of cancer treatment, were shown to have severe uveitis as a side effect, which, although transient and manageable, must be taken into consideration when treating patients [192,193,194,195].

***Lens and Cataract:*** ERK2 is not required for early lens development but it becomes essential for cell proliferation in the germinative zone during the lens growth phase [196]. Oxidation is an essential factor during cataract pathogenesis, and many studies have focused on the investigation of oxidative stress in lens epithelial cells. Thus, nearly all the current knowledge on the association of MAPK pathways and the lens physiology-pathophysiology originates from in vitro studies in lens epithelial cells. Moreover, most of these studies have drawn indirect conclusions, implicating the effector compound that is the main focus of each particular study with cell fate endpoints via the MAPK pathways. For instance, in lens epithelial cells, p-Coumaric acid [197], Honokiol [198] and miR-182-5p [199] have been reported to protect against oxidative stress apoptosis by suppressing the phosphorylation of ERK1/2, JNK, and p38, while Decorin also has been shown to have anti-apoptotic effects, which were imparted by at least p38 suppression [200]. The direct inhibition of p38 by SB203580 in vitro in lens epithelial cells has been shown to be sufficient for imparting protection against oxidative stress-induced apoptosis [201]. In similar in vitro lens epithelial cells studies, Fisetin [202], and grape seed proanthocyanidin extract [203] have been shown to suppress the JNK and p38 pathways leading to protection against oxidative stress damage, while L-carnitine had the same effects by suppressing ERK1/2 and p38 [204]. Conversely, Sanguinarine was reported to upregulate JNK and p38 signaling leading to increased lens epithelial cell apoptosis [205]. Recently, two further in vitro studies reported an amelioration of oxidative stress-related damage by MAPK suppression through regulation of the NLRP3 inflammasome [206] or through as of yet unelucidated functions of the flavonoid Diosmetin from chrysanthemum, that is used in traditional Chinese medicine [207]. Another recent study has similarly reported corroborating results, with p38 and ERK phosphorylation downregulation also being associated with decreased apoptosis via another proxy (TRIM22) [208]. In a single study reporting an apparently opposite effect, Calmodulin like 3 (CALML3) silencing led to increased apoptosis accompanied by a decreased activation of ERK1/2 and JNK [209]. All these studies, however, did not demonstrate a physiologic link to the pathology of cataract, apart from the specific cell type that had been used. In an in vivo study, Taxifolin was reported to act protectively against cataract in a rat model, by suppressing ERK1/2 and p38 [210]. There have also been reports of upregulated ERK1/2 and JNK signaling in induced cataract in rats [211], and a case report of mature cataract in a patient with Noonan syndrome, a genetic gain-of-function RASopathy where, not surprisingly, ERK and p38 were found upregulated in the lens [212]. In light of the absence of sufficient mechanistic in vivo data, however, more research is required for an association between the ERK/JNK/p38 pathways and cataract to be reliably drawn.

### 3.4. MAPK InhibitorAdverse Effects

Much of the current knowledge on the ocular effects of systemic pharmacological intervention in the MAPK signaling pathways has been gained from studies on cancer patients treated with MEK1/2 inhibitors, B-Raf inhibitors or a combination of both [213]. The two main side effects of these therapies are uveitis, in the case of B-Raf inhibitors, and for MEK inhibitors, central serous chorioretinopathy [214], a condition characterized by leakage and accumulation of fluid under the retina (usually in the region of the central macula), retinal detachment and visual impairment (often temporary, with blurred or gray spots in the central visual field). Additionally, vascular injuries have been observed after treatment discontinuation of both B-Raf and MEK inhibitors [215]. The reported adverse effects of specific MAPK inhibitors are summarized in Table 2.

MEK inhibitors can lead to different degrees of retinal, uveal, and adnexal ocular adverse effects, causing visual disturbances or discomfort. One of the most relevant ocular adverse effects of MEK therapy is MEK inhibitor–associated retinopathy (MEKAR). The clinical characteristics of MEKAR include the development of serous detachments of the neurosensory retina. These fluid foci are central serous retinopathy (CSR)-like, but with distinct characteristics such as more commonly being bilateral (92%), multifocal (77% with a median of 6 foci), having both a macular and foveal localization, and a layer depth strictly between the RPE and the interdigitation zone [216]. The clinical picture suggests that effective MEK inhibition interferes with the maintenance of the outer retinal barrier and/or the phagocytic/pump functions of the RPE [217]. MEKAR is usually mild, self-limited, and may subside after continuous use of the drug for weeks or months, or discontinuation, thereby restoring the normal visual function of the retina, with some exceptions [218]. It has been reported, however, that binimetinib, a MEK inhibitor, induces MEKAR with daily fluctuations depending on the time interval after medication, which only partially recovers, being detectable many months later and sometimes leading to subclinical retinal thinning [218]. A minority of patients taking binimetinib also develop mild and transient visual symptoms [219,220]. Other reported features of MEKAR include the thickening of the ellipsoid zone and a “starry sky” pattern of distribution of subretinal granular deposits, with suspected photoreceptor/ RPE toxicity and dysfunction [221]. These effects seem to be function-specific to the MEK inhibition itself and not compound side-effects from alternative actions, as the same clinical observations with the same levels of MEKAR severity have been also observed in patients receiving pimasertib [222] or cobimetinib (a MEK inhibitor), which was manageable in the latter case even without a dose modification, and with no visual loss or permanent retinal damage being detected [223].

Very similar clinical manifestations, but with an increased occurrence of concomitant intraretinal fluid, have also been observed in patients receiving ERK inhibitors, effects that also did not cause irreversible loss of vision or serious eye damage and that did not require medical intervention [224]. It is worth mentioning that these clinical manifestations were observed in patients under ERK inhibitor treatment for metastatic cancer, and not any ocular disease. It is, therefore, unknown what the long-term effects of such treatments would be when targeting populations with specific eye disease.

While MEK inhibitor monotherapy is mainly associated with retinal and choroidal abnormalities, but also with diffuse abnormalities and panuveitis, B-Raf inhibitor monotherapy is significantly associated with iris and ciliary body abnormalities, diffuse abnormalities, and anterior uveitis and panuveitis. Importantly, the combination of these inhibitors shows possible additive side effects [225], with both uveitis and serous retinal detachment having been reported after systemic Dabrafenib (B-Raf inhibitor) and Trametinib (MEK inhibitor) [193]. One report on a combination treatment with Dabrafenib + Trametinib for advanced melanoma documented very frequent subclinical separation of the photoreceptor outer segment from the apical retinal pigment epithelium without inner retinal changes or signs of inflammation [217], while severe bilateral panuveitis during a combined Dabrafenib + Trametinib treatment has been separately reported on multiple occasions [194,226].

**Table 2 cells-12-00617-t002:** Summary of MAPK inhibitors, their target kinases and reported adverse effects.

Inhibitor	Kinase Inhibited	Adverse Effects	Frequency
Binimetinib [213,219,227]	MEK	MEKAR	9–62%
		Subfoveal neurosensory retinal detachment	
		Pigment epithelial detachment	
		Retinal vein occlusion	
		Dry eye	10.0%
		Blurred vision	20.0%
		Visual impairment	
		Retinal thinning	
		Visual symptoms	
Pimasertib [213,222,227,228]	MEK	MEKAR	33–50%
		Serous retinal detachment	10.0%
		Macular degeneration	24%
		Visual field defects	
		Macular edema	
		Retinal vein occlusion	
		Blurred vision	17.0%
		Visual impairment	
Cobimetinib [213,223]	MEK	MEKAR	11.7%
		Floaters	
		Periorbital edema	
		Ocular icterus	
		Blurred vision	4.75–50%
		Visual impairment	
		Increased lacrimation	
		Photophobia	
Trametinib [213]	MEK	MEKAR	1–2%
		Retinal vein occlusion	
		Dry eye	7%
		Eye pain	5.5%
		Increased IOP	
		Conjunctivitis	
Selumetinib [213,229]	MEK	MEKAR	1.0%
		Retinal vein occlusion	1.0%
		Ocular hypertension	15%
		Floaters	
		Epiphora	
		Periorbital edema	
		Blurred vision	30.0%
		Light disturbances	
		RPE detachment	
Refametinib [213]	MEK	MEKAR	
		Retinal vein occlusion	
		Floaters	
		Cataracts	
		Dry eye	
		Blurred vision	
Ulixertinib [224]	ERK	MEKAR	80%
		Multifocal subretinal fluid	
		Blurry vision	20.0%
KO-947 [224]	ERK	Multifocal subretinal fluid	
		Blurry vision	11.0%
		Metamorphopsia	44.0%
Dabrafenib [227,230,231]	b-Raf	Uveitis	1–10%
		Blurred vision	1–10%
		Eye pain	
		Change in color vision	
		Photophobia	
		Eye redness	
		Tearing	

## 4. Conclusions

From this review, it is clear that the MAPK pathways play crucial roles in ocular development and pathophysiology. Each single MAPK pathway exerts functions in diverse physiological processes while conversely, a single physiological process may involve multiple MAPK pathways. This may explain the recent interest in using MAPK inhibitors, alone or in combination with other therapeutics, for the treatment of ocular diseases. In vitro and animal studies have shown positive effects of specific MAPK inhibition in the context of AMD, glaucoma and aniridia; however, caution should be exercised in the use of broad inhibitors of these pathways, given their potential for causing adverse effects in the eye. On the other hand, such effects have been observed only with systemic administration, and in sustained therapeutic schemes that did not specifically aim to monitor and ameliorate eye pathologies. This leaves open for investigation the potential that topical delivery could be safer, and a time-limited targeted intervention could be more effective against specific eye pathologies.

## Figures and Tables

**Figure 1 cells-12-00617-f001:**
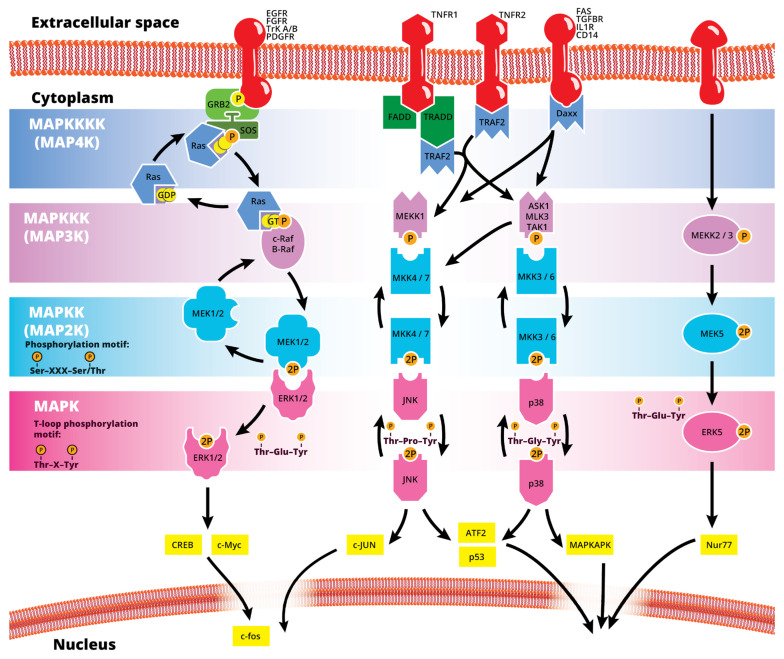
Simplified schematic summary of the main MAPK signaling pathways.

**Figure 2 cells-12-00617-f002:**
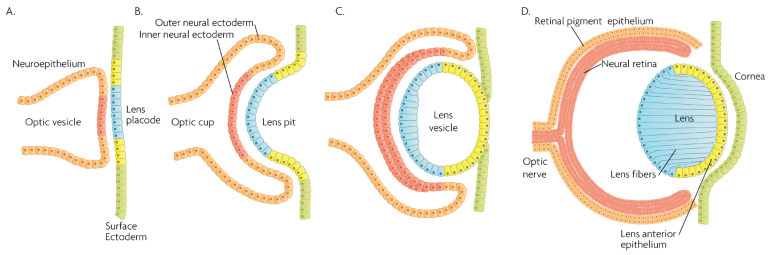
Schematic overview of developmental events during mammalian eye development, and germ layer origin of structures in the eye. (**A**) The optic vesicle, derived from the neuroepithelium of ectodermal lineage, approaches the surface ectoderm where the lens placode (blue cells) forms at the area of proximity between the layers. (**B**) The optic vesicle forms the optic cup, by the concurrent invagination of both the lens placode, forming the lens pit, and the proximal layer of the optic vesicle to the surface ectoderm, forming the presumptive neural retina (red cells). (**C**) The lens pit closes up onto itself forming the lens vesicle, with the cells from the central part of the lens pit (blue cells) directed posteriorly, and the cells from the lens pit periphery (yellow cells) directed anteriorly. The optic cup continues to invaginate. (**D**) The invaginated (inner) layer of the optic cup differentiates into the neural retina (red cells), while the outer layer forms the retinal pigment epithelium, RPE (orange cells). Cells in the posterior surface of the lens vesicle elongate towards the opposite pole, forming the lens fibers and filling the central volume of the lens, while the cells on the anterior side form the lens anterior epithelium. The surface ectoderm closes after the lens vesicle detaches, and the now continuous surface ectoderm forms the cornea.

**Figure 3 cells-12-00617-f003:**
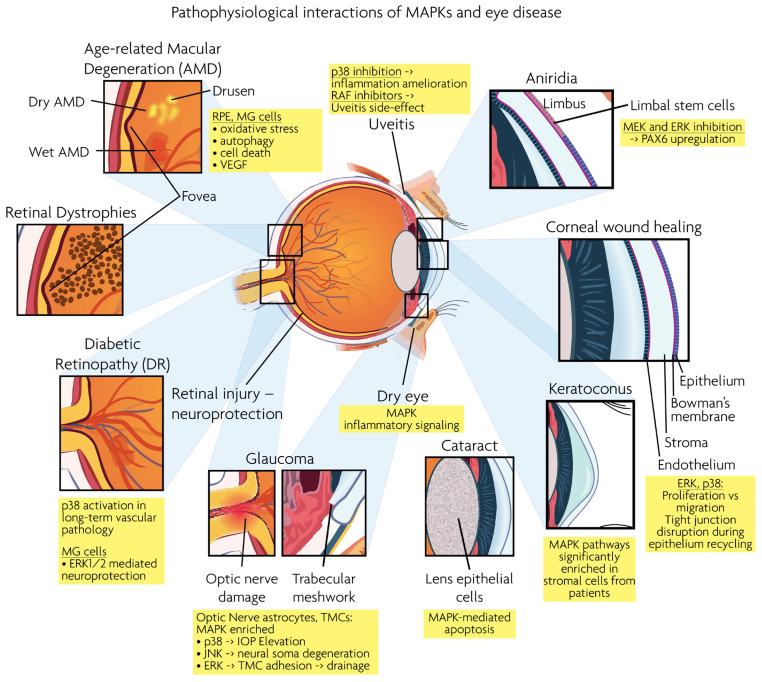
Graphical overview of eye structures and conditions for which MAPK interactions have been reported.

**Table 1 cells-12-00617-t001:** Summary of MAP kinases (up to MAP3Ks) with gene names, protein names and alternative names, the pathways they are known to interact with and their relative level at the signaling cascade.

Gene Name	Protein Name	Alternative Protein Names	Pathway Involved	MAPK Level	Other Gene/Protein Names
MAPK1	ERK2	p42-MAPK	MEK/ERK	MAPK	MAPK2, p38, p40, p41, ERT1, NS13
MAPK3	ERK1	p44-MAPK	MEK/ERK	MAPK	ERT2, PRKM3
MAPK4	ERK4	p63-MAPK	atypical MAPK	MAPK	PRKM4
MAPK6	ERK3	p97-MAPK	atypical MAPK	MAPK	PRKM6, HsT17250
MAPK7	ERK5		ERK5	MAPK	PRKM7, BMK1
MAPK8	JNK1	SAPK1	JNK	MAPK	PRKM8
MAPK9	JNK2	p54aSAPK	JNK	MAPK	PRKM9
MAPK10	JNK3	p54bSAPK	JNK	MAPK	PRKM10, SAPK1b, p493F12
MAPK11	p38 beta	SAPK2, SAPK2B	p38	MAPK	PRKM11
MAPK12	p38 gamma	ERK6, SAPK-3	p38	MAPK	PRKM12
MAPK13	p38 delta	SAPK4	p38	MAPK	PRKM13
MAPK14	p38 alpha	SAPK2A, Mxi2	p38	MAPK	PRKM14, PRKM15, CSBP, EXIP
MAPK15	ERK7/8		atypical MAPK	MAPK	
MAP2K1	MEK1	MKK1, MAPKK1	MEK/ERK	MAP2K	CFC3
MAP2K2	MEK2	MKK2, MAPKK2	MEK/ERK	MAP2K	CFC4
MAP2K3	MEK3	MKK3, MAPKK3	p38	MAP2K	SAPKK2
MAP2K4	MEK4	MKK4, MAPKK4	JNK	MAP2K	SAPKK1, JNKK1, JNKK
MAP2K5	MEK5	MAPKK5	ERK5	MAP2K	
MAP2K6	MEK6	MKK6, MAPKK6	p38	MAP2K	SAPKK3
MAP2K7	MEK7	MKK7, MAPKK7	JNK	MAP2K	SAPKK4, JNKK2
RAF1	c-Raf	Raf-1	MEK/ERK	MAP3K	
BRAF	B-Raf	BRAF-1, RAFB1	MEK/ERK	MAP3K	NS7
MAP3K1	MEKK1		JNK	MAP3K	
MAP3K2	MEKK2	MEKK2B	ERK5	MAP3K	
MAP3K3	MEKK3	MAPKKK3	ERK5	MAP3K	
MAP3K4	MEKK4	MAPKKK4		MAP3K	MTK1, PRO0412
MAP3K5	ASK1	MEKK5, MAPKKK5	JNK and p38	MAP3K	
MAP3K6	ASK2	MEKK6, MAPKKK6		MAP3K	
MAP3K7	TAK1	MEKK7, TGF1a	JNK and p38	MAP3K	CSCF, FMD2
MAP3K8	MEKK8	Tpl-2, c-COT		MAP3K	COT, EST, ESTF, AURA2
MAP3K9	MLK1	MEKK9		MAP3K	PRKE1
MAP3K10	MLK2	MEKK10		MAP3K	MST
MAP3K11	MLK3	MEKK11	JNK and p38	MAP3K	PTK1, SPRK
MAP3K12	ZPK	MEKK12		MAP3K	DLK, MUK, HP09298
MAP3K13	LZK	MEKK13	JNK	MAP3K	MLK
MAP3K14				MAP3K	FTDCR1B, HS, HSNIK, NIK
MAP3K15	ASK3			MAP3K	bA723P2.3
TAOK1	PSK2	MAP3K16, TAO1	JNK	MAP3K	DDIB, KFC-B, MARKK, hKFC-B
TAOK2	PSK	MAP3K17, TAO2		MAP3K	Tao2beta, PSK1-BETA
TAOK3		MAP3K18	p38	MAP3K	DPK, JIK, hKFC-A
MAP3K19				MAP3K	RCK, YSK4
MAP3K20	MLK7	mlklak, pk		MAP3K	AZK, MLT, MRK, ZAK, SFMMP
MAP3K21	MLK4	dJ862P8.3		MAP3K	

## Data Availability

No new data were created.

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
