# Peer review of "MAPK Pathways in Ocular Pathophysiology: Potential Therapeutic Drugs and Challenges"

_cells, 2023, doi:10.3390/cells12040617_

Round 1

Reviewer 1 Report

No Comments

Author Response

Dear editor,

We have now completed the revision of our manuscript. During the revision process, we have individually addressed all the comments made by the reviewers. As a result, we believe that the submitted manuscript has been substantially improved. Below we submit our reply to the reviewers, where we reply to each comment on a point-by-point basis, describing the changes made on the main manuscript that are relevant to that comment, justify our decisions on certain subjects and discuss further, when needed. Because the inbuilt “track-changes” feature in the journal-provided MS word document had the side effect of inconsistent line numbering depending on the display mode for changes, we have instead tracked the changes manually with red text in the enclosed revised manuscript.

Apart from the changes that are tracked on the submitted manuscript and referenced to our replies below, please note three additional changes made in the revision process:

  • Figure 1 has been updated with slight visual changes, and also the caption of “Raf-1” has been changed to “c-Raf / B-Raf”, so that consistent naming is throughout the manuscript, figures and tables.
  • Figure 3 (previously figure 2) has been slightly visually updated.

  • Since reviewer #2 requested reference re-organization and additions, it was necessary to re-introduce in the manuscript the references as citation links, rather than a stripped plain text that the journal provided back to us. This has been done without tracking to the entirety of the manuscript, not to overwhelm the text with changes. Updated reference numbers throughout the manuscript do not appear as changes. The old, "plain text" references section has been deleted and replaced by new dynamic links that bear the same text.

Regarding all figures in the manuscript, we currently provide them as embedded MS Word objects to facilitate the review process, but publication and print quality versions of them are available to you for the eventual typesetting process, should you decide to accept our manuscript.

We thank you for your consideration of our manuscript, and look forward to your and the reviewer’s feedback on our revised manuscript.

Reviewer #1

No Comments

We thank Reviewer #1 for critically reviewing our manuscript and for giving us a positive assessment in the reviewing process.

Reviewer 2 Report

In their manuscript, the authors describe the several involvements of MAPK in the eye. While this attempt is praiseworthy, the manuscript needs a thorough revision before publication can be considered. It is rather confusing to read.

The authors need to introduce in the physiology and development of the eye before describing the role of the MAPK there. Otherwise it is not really understandable to the reader.

Page 2, line 83 (“… but no instance of …”, I do not understand the sentence.

Please explain the RPE before writing about the role of MAPK in RPE. Please explain why their role in proliferation is of any importance considering that the RPE is post mitotic in the adult.

When describing the role of MAPK in retinal regeneration in newt and zebrafish, it would be of importance to discuss whether this has any relevance to the human situation, as mammals do not have the capacity of retinal regeneration.

MAPK in the retina, the authors are missing out on several aspects that have been shown of ERK MAPK involvement in RPE cell death and VEGF regulation, e.g. its regulation under oxidative stress. Furthermore, the authors cite reviews (e.g. reference 57 or 60), instead of the original papers. Please revise this chapter accordingly and change the citation to the original papers. Please revise the whole manuscript, replacing reviews with original papers.

Retinal dystrophies, “… group of diseases caused by mutations in multiple genes”, this is not very precise. Please give a more distinguished description of retinitis pigmentosa and its causes.

Aniridea, please note that Pax6 haploinsufficiency can also cause another disease entity, called Peter’s anomaly. How can the effect of an “ocular surface disease” be shown in a mouse’s retina (line 448)? Please explain. Is AAK the same as ARK? (line 459)?

Which MAPK is MAPK1? Please stay to one nomenclature. A table of the different MAPK and MAPKK (and MAPKKK) would be extremely helpful for understanding of the text. In addition, the authors speak of many inhibitors, please also include a table of the names of these inhibitors and their targets. In addition, the authors are jumping also to different other treatments and inhibitors, whose connection to MAPK is not always clear to the reader (for example B-raf inhibitors). Please always explain the relevance of all the mentioned inhibitors to MAPK.

Page 13, first paragraph, the authors state an “absence of sufficient in vivo data”, while in the sentence before, they quote a rat study?

Central serous chorioretinopathy, please give a brief explanation as to what this condition is.

Please elaborate more in the actual clinical potential in the different conditions. For which conditions would a MAPK inhibitor really suitable? For which not?

Author Response

Dear editor,

We have now completed the revision of our manuscript. During the revision process, we have individually addressed all the comments made by the reviewers. As a result, we believe that the submitted manuscript has been substantially improved. Below we submit our reply to the reviewers, where we reply to each comment on a point-by-point basis, describing the changes made on the main manuscript that are relevant to that comment, justify our decisions on certain subjects and discuss further, when needed. Because the inbuilt “track-changes” feature in the journal-provided MS word document had the side effect of inconsistent line numbering depending on the display mode for changes, we have instead tracked the changes manually with red text in the enclosed revised manuscript.

Apart from the changes that are tracked on the submitted manuscript and referenced to our replies below, please note three additional changes made in the revision process:

  • Figure 1 has been updated with slight visual changes, and also the caption of “Raf-1” has been changed to “c-Raf / B-Raf”, so that consistent naming is throughout the manuscript, figures and tables.
  • Figure 3 (previously figure 2) has been slightly visually updated.

  • Since reviewer #2 requested reference re-organization and additions, it was necessary to re-introduce in the manuscript the references as citation links, rather than a stripped plain text that the journal provided back to us. This has been done without tracking to the entirety of the manuscript, not to overwhelm the text with changes. Updated reference numbers throughout the manuscript do not appear as changes. The old, "plain text" references section has been deleted and replaced by new dynamic links that bear the same text.

Regarding all figures in the manuscript, we currently provide them as embedded MS Word objects to facilitate the review process, but publication and print quality versions of them are available to you for the eventual typesetting process, should you decide to accept our manuscript.

We thank you for your consideration of our manuscript, and look forward to your and the reviewer’s feedback on our revised manuscript.

Reviewer #1

No Comments

We thank Reviewer #1 for critically reviewing our manuscript and for giving us a positive assessment in the reviewing process.

Reviewer #2

In their manuscript, the authors describe the several involvements of MAPK in the eye. While this attempt is praiseworthy, the manuscript needs a thorough revision before publication can be considered. It is rather confusing to read.

We thank the reviewer for critically reviewing of our manuscript and for the valuable feedback. We believe that the revised manuscript, following the reviewer’s comments is an improved one, with a clearer structure and flow for the reader. Below are our responses and a description of the changes made in the manuscript, in a point-by-point manner.

The authors need to introduce in the physiology and development of the eye before describing the role of the MAPK there. Otherwise it is not really understandable to the reader.

To address the reviewer’s comment, we have added a paragraph in the introduction, briefly describing the structures of the eye that are relevant to this manuscript, and their function (lines 66-101). We have also added a new figure (Figure 2) to visually introduce the development of the eye and the developmental cell lineages. New Figure 2 also describes the major structures of the eye that are discussed in this manuscript (neural retina, retina pigment epithelium, lens, cornea, optic nerve). In combination with Figure 3 (previously Figure 2), where all the relevant structures of the developed eye that are discussed in our manuscript are visually presented, we now believe that the readers can get a good overview of the physiology and development of the eye.

Page 2, line 83 (“… but no instance of …”, I do not understand the sentence.

The reviewer correctly identified some poor phrasing there. We intended to convey the message that even though random mutations should more often lead to defective proteins and thus to a pathway disruption, all the known mutational defects that relate to the RAS-MAPK pathway are rarer gain-of-function mutations that lead to inefficient pathway inhibition. Thus, we conclude that nonsense mutations, when they occur, must be non-viable. This signifies the importance of the MAPK pathway for the organism’s development. We have rephrased the sentence, and believe that now this point is more clearly conveyed (lines 122-126).

Please explain the RPE before writing about the role of MAPK in RPE. Please explain why their role in proliferation is of any importance considering that the RPE is post mitotic in the adult.

As part of addressing the reviewer’s first comment, we have described the RPE, its function and its development both at the revised introduction (lines 94-97), and via the new Figure 2. RPE proliferation is important and relevant during eye development and also in stem cell research. Following the reviewer’s suggestion, we have briefly commented this issue in the revised manuscript (lines 132-134).

When describing the role of MAPK in retinal regeneration in newt and zebrafish, it would be of importance to discuss whether this has any relevance to the human situation, as mammals do not have the capacity of retinal regeneration.

Indeed, this discussion was missing from our manuscript, and we thank the reviewer for pointing this out. We believe that by addressing the reviewer’s previous comment on the RPE proliferation (lines 132-134), we have partly addressed this comment. We have additionally expanded the “Physiological role of MAPKs in the eye” section to clarify that the retinal physiology of teleost fish is distinctly different of that in mammals, as fish have the innate ability for retinal regeneration after injury (lines 143-147). Knowledge on the fish and amphibian retinal physiology is not totally irrelevant though, since it can be applied to stem cell research for the induction of pluripotent cells to differentiate to RPE cells, in potential research for induced retinal regeneration with pharmacological intervention, or even for developmental research purposes. This has now been also discussed in the revised manuscript (lines 143-147).

MAPK in the retina, the authors are missing out on several aspects that have been shown of ERK MAPK involvement in RPE cell death and VEGF regulation, e.g. its regulation under oxidative stress. Furthermore, the authors cite reviews (e.g. reference 57 or 60), instead of the original papers. Please revise this chapter accordingly and change the citation to the original papers. Please revise the whole manuscript, replacing reviews with original papers.

We thank the reviewer for bringing this to our attention. Indeed, although we had covered the role of VEGF and MAPK interactions both in the cases of AMD and in the section of diabetic retinopathy, a direct discussion on RPE cell death, VEGF, MAPKs and oxidative stress was missing. We have now expanded this section to include information on the interplay between MAPKs and VEGF under oxidative stress in the RPE (lines 224-233, 237-238). Also, we have included one additional study connecting high-glucose induced stress with VEGF expression, p38 signaling and apoptosis in RPE cells, under the Diabetic retinopathy section (lines 293-294).

Regarding references 57 and 60 (references 61 and 65, respectively, in the revised manuscript), we contend that they have been appropriately used in our manuscript because both are cited for conclusions drawn from the revision and study of multiple publications. As such, it is the original conclusions of the cited authors that we are correctly attributing. If we opted instead to cite the multiple studies these conclusions were drawn upon, it would be disingenuous of us to allure that we are the ones that drew the conclusion. In particular, old reference 57 (ref. 61 in the revised manuscript) is cited for the following conclusion: “it seems that this dual role is dependent on signal duration, with short-term ERK1/2 signaling being pro-proliferative and inversely, persistent ERK1/2 activation leading to cell death”, referring to reviewed studies with short duration, and other with longer signaling duration. Old Reference 60 (ref. 65 in the revised manuscript) is cited for the statement “MAPK signaling pathways are the most widely used in initiating the release of cytokines and chemokines in RPE cells” which is a statement that obviously cannot be drawn from any single study.

Following the reviewer’s suggestion, we have gone through all the references to ensure proper usage throughout the manuscript. As a result of this reorganization of references, we have included two original studies to complement reference 61, as per the reviewer’s request, we have removed two review articles where we found the claims in the manuscript were not properly supported by the cited literature (ref. 78 and 155 of the old manuscript) and we have also added two original research citations (ref. 78 and ref. 159).

Retinal dystrophies, “… group of diseases caused by mutations in multiple genes”, this is not very precise. Please give a more distinguished description of retinitis pigmentosa and its causes.

We thank the reviewer for this suggestion. We have now provided a more detailed description of RP in out revised manuscript (lines 309-312).

Anirideae, please note that Pax6 haploinsufficiency can also cause another disease entity, called Peter’s anomaly. How can the effect of an “ocular surface disease” be shown in a mouse’s retina (line 448)?

The reviewer correctly identified that our previous description of aniridia was insufficient. Aniridia is a panocular disease affecting the development and function of multiple parts of the eye. In our previous manuscript version though, AAK, which is a common manifestation of aniridia could be confused with the disease itself, leading to the confusion of why we opted to include research on the retina in our discussion of aniridia. We have now revised this section, clarifying that conditions like Peters’ anomaly or AAK are defects that are common manifestations the disease, but the disease itself has a heterogeneous clinical image and may present with panocular manifestations (lines 482-495). We think that the revised section is clearer on these distinctions now, and justifies our discussion of the long term results in the retina by Cole J. et al., 2019 (ref. 173). We have also expanded on the specific results of this study, to avoid confusion.

Please explain. Is AAK the same as ARK? (line 459)?

Indeed, AAK stands for Aniridia Associated Keratopathy while ARK stands for Aniridia Related Keratopathy. Since both denote the same condition, we switched into using AAK consistently in the manuscript.

Which MAPK is MAPK1? Please stay to one nomenclature. A table of the different MAPK and MAPKK (and MAPKKK) would be extremely helpful for understanding of the text. In addition, the authors speak of many inhibitors, please also include a table of the names of these inhibitors and their targets. In addition, the authors are jumping also to different other treatments and inhibitors, whose connection to MAPK is not always clear to the reader (for example B-raf inhibitors). Please always explain the relevance of all the mentioned inhibitors to MAPK.

We thank the reviewer for identifying a discrepancy in the nomenclature use in our manuscript. MAPK1 is ERK2. For consistency, we have updated the name used in the manuscript (line 553). As per the reviewer’s suggestions, we have also updated the manuscript with a new table (Table 1), summarizing the different MAPKs, alternative names and their positions at the signaling pathways, to complement Figure 1 that served a similar purpose in a visual manner. Since all inhibitor targets that are mentioned in this manuscript are contained in this new table and in Figure 1, we think that the revised manuscript is now much easier to follow with respect to the pathways that each inhibitor affects. Regarding the reviewer’s suggestion about a table with the various inhibitors and the inhibited target, we believe that Table 1 of the original manuscript (renamed to “Table 2” in the revised manuscript) already served this purpose. We have updated the table’s caption to be more descriptive of that fact.

Page 13, first paragraph, the authors state an “absence of sufficient in vivo data”, while in the sentence before, they quote a rat study?

Indeed. We do not think there is a contradiction there. We do not claim total absence of data, we make the assessment that there are insufficient data for safe conclusions to be drawn, and then assert that more research is needed for that. To our knowledge, the cited studies in the revised manuscript are the only ones with in vivo data in the literature. Following the reviewer’s comment, we did a more extensive literature search for in vivo studies that we might have missed. We have updated the manuscript with two additional studies (ref. 211-212) with observational data, and have rephrased slightly (lines 582-585).

Central serous chorioretinopathy, please give a brief explanation as to what this condition is.

We thank the reviewer for identifying our omission in explaining this condition. This has now been addressed in the revised manuscript (lines 593-597).

Please elaborate more in the actual clinical potential in the different conditions. For which conditions would a MAPK inhibitor really suitable? For which not?

Unfortunately, current knowledge on the clinical potential of pharmacological MAPK inhibition by selective inhibitors for treating ocular disease is very limited. MAPK inhibitors have been used in some in vitro and animal studies to elucidate pathway functions and interactions, or to investigate potential therapeutic opportunities. When this is the case, it is discussed in the context of the particular disease model for AMD, for glaucoma and for aniridia in the respective sections. Apart from that, all knowledge of clinical relevance is limited to reported ocular side effects and case reports, from patients under cancer chemotherapy. The authors feel that the section specifically illustrating ocular adverse effects of MEK inhibitors is not the appropriate place to discuss the clinical potential of the same compounds. That being said, it must be noted that the clinical knowledge on the effects of these inhibitors comes from systemic administration and over sustained therapies that did not specifically aim to ameliorate eye pathologies. It is unknown whether effects of clinical significance could be observed in targeted interventions and topical administration. We have expanded our conclusion section, where this was discussed in a general way, apart from the specific discussions in each respective disease section.

Round 2

Reviewer 2 Report

The authors have answered all this reviewer's comments to the reviewer's satisfaction. Indeed, a very fine paper on this subject!